# High-Fidelity Harmonic Generation in Optical Micro-Resonators Using BFGS Algorithm

**DOI:** 10.3390/mi11070686

**Published:** 2020-07-15

**Authors:** Özüm Emre Aşırım, Alim Yolalmaz, Mustafa Kuzuoğlu

**Affiliations:** 1Department of Electrical and Electronics Engineering, Middle East Technical University, 06800 Ankara, Turkey; kuzuoglu@metu.edu.tr; 2Micro and Nanotechnology Program, Middle East Technical University, 06800 Ankara, Turkey; alim.yolalmaz@metu.edu.tr

**Keywords:** harmonic generation, non-linear wave mixing, non-linear programming, micro-resonator

## Abstract

Harmonic generation is an attractive research field that finds a variety of application areas. However, harmonic generation within a medium of micron-scale interaction length limits the magnitude of nonlinear coupling and leads to poor harmonic generation efficiency. In this study, we present a constrained non-linear programming approach based on the Quasi-Newton Broyden–Fletcher–Goldfarb–Shanno (BFGS) algorithm to obtain high-fidelity harmonic generation in optical micro-resonators. Using this approach, one can achieve high-intensity harmonic generation in a simple Fabry–Perot type optical micro-resonator. The generation of super-intense harmonics at a typical ultraviolet (UV)-ablation frequency of 820 THz and at pure yellow-light (515 THz) is investigated in particular. Moreover, we achieved more than 98% accuracy compared to well-known theoretical results. Our approach enables the design of highly efficient microscale harmonic generators to be used in integrated photonic devices.

## 1. Introduction

Microscale harmonic generation is of high interest for many modern technologies, including optical antennas, lab-on-chip devices, and integrated optical devices. Currently, the conversion efficiency of microscale optical harmonic generators is very low due to the very small interaction length. Highly efficient targeted harmonic generation can be achieved with interaction mediums that are at least a few centimeters long. However, theoretically, the efficiency of generating a desired harmonic can be increased in the microscale by increasing the intensities of the waves that are intermixed, such an increase in intensity will definitely damage the micron-sized optical medium. Another solution may be to use an interaction material that has a superior nonlinearity, but materials with superior nonlinearity are usually not available. Even in the case of availability, super-nonlinear materials can be quite costly to employ in micron size as they are usually fabricated in thin film form and need to be arranged as an array of thin films to achieve the desired conversion efficiency. The discovery of novel super-nonlinear materials via experimentation is a hot-topic and some artificial materials do display an unusually high nonlinearity [1,2,3,4,5].

Although highly nonlinear natural materials and super-nonlinear artificial materials may help to increase the harmonic generation efficiency in the microscale, these materials usually lead to high dielectric, conduction, and scattering losses, along with being too expensive to afford or to fabricate. For these reasons, high-efficiency harmonic generation remains as an issue in the microscale. In this study, we used a computational approach to tackle this issue. A computational study that is based on maximizing the harmonic generation efficiency via a nonlinear programming algorithm is lacking [6,7,8]. Experimental studies usually focus on finding new techniques to increase the second and the third harmonic generation efficiency and there are relatively few experimental studies that have demonstrated a minor increase in the harmonic generation or conversion efficiency via certain experimental configurations and setups [9,10,11,12,13,14,15,16,17,18,19,20,21,22,23,24,25,26,27,28,29,30,31,32,33,34,35,36,37,38,39]. Some computational studies have focused on increasing the computation efficiency of harmonic generation problems rather than proving that the harmonic generation efficiency itself can actually be increased. These include the studies mentioned in [6,7,8], which have managed to increase the efficiency of the nonlinear Finite-Difference-Time-Domain (FDTD) method by decreasing the dispersion error. The current literature of harmonic generation via nonlinear wave mixing is dominated by increasing the efficiency of the second/third harmonic generation rather than increasing the efficiency of an arbitrary harmonic [9,10,11,12,13,14,15,16,17,18,19,20,21,22,23,24,25,26,27,28,29,30,31,32,33,34,35,36,37,38,39]. There are currently no known techniques in the literature that reported a high-efficiency in the microscale for an arbitrary harmonic generation (not necessarily the second harmonic) under monochromatic optical excitation [9,10,11,12,13,14,15,30,31,32,33,34,35,36,37,38,39]. In this study, we will show that for an arbitrary excitation frequency, one can maximize the harmonic generation efficiency at an arbitrary frequency. For example, if a microcavity is excited with an infra-red pump wave, and we want a super-high harmonic generation efficiency in the ultraviolet spectral range, we will show that this is possible by using constrained non-linear programming. The generation of a desired harmonic with an ultra-high efficiency is important for many applications in optics and biomedical engineering, such as for integrated optical devices that are used in spectroscopy and for Lab-On-Chip devices that are used for medical diagnostics. Another research field that can benefit from highly efficient harmonic generation at a desired frequency is the field of optical antennas. Optical antennas that can generate light at an arbitrary frequency would become ultra-wideband optical antennas through the embedding of a microscale controller. The microscale controller would have to do what we aim to do in this study, the adjustment of the micro-resonator parameters for maximizing the harmonic generation efficiency. Most importantly, ultra-efficient microscale harmonic generators can pave the way for macroscale high-efficiency harmonic generators that can generate powerful THz rays, ultraviolet (UV) rays, X-rays, and even gamma rays with much higher efficiencies than are currently available. High power yellow-light and ultraviolet-light generation is of particular interest as the power output of many yellow-light and ultraviolet-light generators is quite low. Especially high intensity UV-harmonic generation is an important issue as the harmonic conversion efficiency at this level is extremely low [40,41,42,43,44,45]. To provide a mathematical and computational proof that high-efficiency harmonic generation is feasible in a micro-resonator, we will present a full-fledged dispersion analysis. We will first investigate the nonlinear wave propagation concept in an arbitrary multi-resonant medium that is placed as an interaction medium in a plain Fabry–Perot type micro-resonator. Then, we will define the wave equation and its associated polarization density equations for each resonance frequency. Finally, we will define the pump wave as a combination of M ultrashort pulses and we will tune the pulses of these high-intensity pulses to maximize the frequency conversion efficiency at a target frequency that is to be generated. We will achieve this through an efficient constrained nonlinear programming algorithm that has a relatively lower computation cost and a relatively faster convergence rate such as the Quasi-Newton type Broyden–Fletcher–Goldfarb–Shanno (BFGS) algorithm. We will start our analysis by presenting the partial differential equations involved in nonlinear wave propagation in a multi-resonant interaction medium placed in a micro-resonator, and then we will present the formulations for the BFGS algorithm for the frequency tuning of the intense pulses that form the source wave which energizes the micro-resonator.

## 2. Nonlinear Wave Interaction in Optical Micro-Resonators

The interaction medium of an optical micro-resonator may have a single dominant resonance (emission) frequency or it may have multiple resonance frequencies associated with different resonance probabilities (oscillator strengths) [40,41,42,43,44,45]. Some materials, such as excitonic materials, have a single resonance frequency, while most materials have more than one resonance frequency. In this study, we will consider an arbitrary medium which has N resonance frequencies associated with N corresponding polarization damping rates. Quantum mechanics dictates that each electron that oscillates at a certain resonance frequency is associated with a resonance probability or oscillator strength and the sum of all resonance probabilities is equal to one. Assuming we have M different waves present in an arbitrary medium, if at least one of the waves has a sufficiently high intensity, then there will be a nonlinear coupling between the M waves and energy transfer between the waves will occur. For the total high-intensity wave that is present in a medium with N resonances, the wave equation that represents the time variation of the electric field E of the high intensity wave and the associated equations that represent the components of the polarization density induced by the high intensity wave are given as
(1)$$\nabla^{2}\left(E\right)-\mu_{0}\varepsilon_{\infty }\frac{\partial^{2}\left(E\right)}{\partial t^{2}}=\mu_{0}\sigma \frac{\partial \left(E\right)}{\partial t}+\mu_{0}\frac{d^{2}P}{\partial t^{2}}$$
(2)$$\frac{d^{2}P_{1}}{dt^{2}}+\gamma_{1}\frac{dP_{1}}{dt}+\omega_{1}{}^{2}P_{1}-\frac{\omega_{1}{}^{2}P_{1}{}^{2}}{Q_{1}ed}+\frac{\omega_{1}{}^{2}P_{1}{}^{3}}{Q_{1}{}^{2}e^{2}d^{2}}=\frac{Q_{1}e^{2}E}{m}$$
(3)$$\frac{d^{2}P_{2}}{dt^{2}}+\gamma_{2}\frac{dP_{2}}{dt}+\omega_{2}{}^{2}P_{2}-\frac{\omega_{2}{}^{2}P_{2}{}^{2}}{Q_{2}ed}+\frac{\omega_{2}{}^{2}P_{2}{}^{3}}{Q_{2}{}^{2}e^{2}d^{2}}=\frac{Q_{2}e^{2}E}{m}$$
(4)$$\frac{d^{2}P_{N}}{dt^{2}}+\gamma_{N}\frac{dP_{N}}{dt}+\omega_{N}{}^{2}P_{N}-\frac{\omega_{N}{}^{2}P_{N}{}^{2}}{Q_{N}ed}+\frac{\omega_{N}{}^{2}P_{N}{}^{3}}{Q_{N}{}^{2}e^{2}d^{2}}=\frac{Q_{N}e^{2}E}{m}$$
where *P* is the total polarization density, *P_i_* is the polarization density component at the ith resonance frequency, *γ*_i_ is the polarization damping rate associated with the ith resonance frequency, *ω*_i_ is the ith angular resonance frequency, *e* is the electron charge, *d* is the atomic diameter, *σ* is the medium conductivity, and $\varepsilon_{\infty }$ is the background permittivity. The number of electrons oscillating at each resonance is indicated by $Q_{i}$ and they are related to the number of electrons per unit volume (electron density) Q via the oscillation strength $\xi_{i}$ such that
(5)$$\overset{N}{\underset{i=1}{\sum }}\xi_{i}=1,\;\;Q_{i}=\xi_{i}Q,\;\;P=\overset{N}{\underset{i=1}{\sum }}P_{i}=\overset{N}{\underset{i=1}{\sum }}Q_{i}p_{i}=Q\overset{N}{\underset{i=1}{\sum }}\xi_{i}p_{i}$$
where, *p_i_* is the dipole moment at the ith resonance.

Therefore, in order to determine the time variation of the electric field, we had to solve these N+1 equations. However, since our goal was to maximize the harmonic generation efficiency, we dealt with the nonlinear programming of these equations to maximize the spectral intensity at the desired frequency.

Assuming that we excite the optical microcavity with M high-intensity ultra-short pulses. The total wave that represents the overall excitation is stated as
(6)$$E\left(x=0\;\mu m,t\right)=\sum_{i=1}^{M}A_{i}\cos\left(2\pi \nu_{i}t+\psi_{i}\right)\left(u\left(t\right)-u\left(t-\Delta T_{i}\right)\right).$$

We wanted to tune the frequencies of the excitation, since there were N+1 equations and M optimization parameters involved in this configuration, we needed a cost-efficient optimization algorithm. The Quasi-Newton type Broyden–Fletcher–Goldfarb–Shanno (BFGS) algorithm enabled a relatively fast convergence at a lower computational cost as the Hessin matrix was recursively updated and the computation of the second derivatives were not needed. Hence, we employed the BFGS algorithm for the optimization part.

Given the expression for excitation in Equation (6), and Equations (1)–(5) that modeled its propagation in the microcavity, we could then define the optimization problem. Note that this was a constrained optimization problem as the source device that generates the excitation could generate the ultrashort pulses at a certain frequency range and at a certain maximum pulse energy. Assuming that the we were solving the problem for a given pulse energy that was below the maximum available pulse energy, we had the allowable frequency range as the constraint of the problem. Most importantly, since we wanted a high-intensity output at a target frequency (desired harmonic), we had to define the cost function carefully and accurately. The intended description for the cost function can be stated as
(7)$$F\left(\nu \right)=\left|E\left(\nu =\nu_{target}\right)\right|=\left|\int_{\nu_{target}-∆\nu }^{\nu_{target}+∆\nu }\left\{\int_{0}^{∆T}\{E\left(x=x^{\prime },t\right)e^{-i\left(2\pi \Omega \right)t}\}dt\right\}e^{i\left(2\pi \Omega \right)t}d\Omega \right|$$
where ∆ν is the bandwidth around the target frequency. A certain target harmonic can be programmed to be maximized because the process of nonlinear wave mixing generates many new harmonics as a result of supercontinuum generation [20,21,22,23,24,25,26], which arise from high stored energy. What we intended to do is to program the microcavity to concentrate the supercontinuum spectral density around the target frequency. Therefore, the summary of the problem can be stated as follows: optimization parameters: $\nu =\left[\nu_{1},\;\nu_{2},\;\ldots ,\nu_{M}\right]$, cost function to be maximized: $F\left(\nu \right)=\left|E\left(\nu =\nu_{target}\right)\right|$, constraints: $\nu_{min}\leq \nu \leq \nu_{max}$, equations: (8)$$\nabla^{2}\left(E\left(\nu \right)\right)-\mu_{0}\varepsilon_{\infty }\frac{\partial^{2}\left(E\left(\nu \right)\right)}{\partial t^{2}}=\mu_{0}\sigma \frac{\partial \left(E\left(\nu \right)\right)}{\partial t}+\mu_{0}\frac{d^{2}P}{\partial t^{2}}$$
(9)$$\frac{d^{2}P_{1}}{dt^{2}}+\gamma_{1}\frac{dP_{1}}{dt}+\omega_{1}{}^{2}P_{1}-\frac{\omega_{1}{}^{2}P_{1}{}^{2}}{N_{1}ed}+\frac{\omega_{1}{}^{2}P_{1}{}^{3}}{N_{1}{}^{2}e^{2}d^{2}}=\frac{N_{1}e^{2}E\left(\nu \right)}{m}$$
(10)$$\;\frac{d^{2}P_{2}}{dt^{2}}+\gamma_{2}\frac{dP_{2}}{dt}+\omega_{2}{}^{2}P_{2}-\frac{\omega_{2}{}^{2}P_{2}{}^{2}}{N_{2}ed}+\frac{\omega_{2}{}^{2}P_{2}{}^{3}}{N_{2}{}^{2}e^{2}d^{2}}=\frac{N_{2}e^{2}E\left(\nu \right)}{m}$$
(11)$$\frac{d^{2}P_{N}}{dt^{2}}+\gamma_{N}\frac{dP_{N}}{dt}+\omega_{N}{}^{2}P_{N}-\frac{\omega_{N}{}^{2}P_{N}{}^{2}}{N_{N}ed}+\frac{\omega_{N}{}^{2}P_{N}{}^{3}}{N_{N}{}^{2}e^{2}d^{2}}=\frac{N_{N}e^{2}E\left(\nu \right)}{m}$$

Note that even though the total excitation wave initially had M frequency components as seen in Figure 1, after the desired interaction duration, the spectrum of the excitation changed due to the nonlinear interaction. What we were trying to achieve here is to maximize the spectral density of the output spectrum around the target frequency, so that the output intensity at the desired harmonic was maximized.

Equations (1)–(4) can be discretized using the FDTD method at each iteration (*k*) of the optimization as follows:(12)$$\begin{array}{l} \frac{E_{k}\left(i+1,j\right)-2E_{k}\left(i,j\right)+E_{k}\left(i-1,j\right)}{∆x^{2}}-{µ}_{0}\varepsilon_{\infty }\left(i,j\right)\frac{E_{k}\left(i,j+1\right)-2E_{k}\left(i,j\right)+E_{k}\left(i,j-1\right)}{∆t^{2}}\\ \;={µ}_{0}\sigma \left(i,j\right)\frac{E_{k}\left(i,j\right)-E_{k}\left(i,j-1\right)}{∆t}\\ \;+{µ}_{0}\frac{P_{k}\left(i,j+1\right)-2P_{k}\left(i,j\right)+P_{k}\left(i,j-1\right)}{\Delta t^{2}} \end{array}$$
(13)$$\begin{array}{l} \;\frac{P_{1,k}\left(i,j+1\right)-2P_{1,k}\left(i,j\right)+P_{1,k}\left(i,j-1\right)}{∆t^{2}}+\gamma_{1}\frac{P_{1,k}\left(i,j\right)-P_{1,k}\left(i,j-1\right)}{∆t}\\ \;+4\pi^{2}f_{1}{}^{2}\left(P_{1,k}\left(i,j\right)\right)-\frac{4\pi^{2}f_{1}{}^{2}}{Q_{1}ed}{\left(P_{1,k}\left(i,j\right)\right)}^{2}+\frac{4\pi^{2}f_{1}{}^{2}}{Q_{1}{}^{2}e^{2}d^{2}}{\left(P_{1,k}\left(i,j\right)\right)}^{3}\\ \;=\frac{Q_{1}e^{2}}{m}\left(E_{k}\left(i,j\right)\right)\; \end{array}$$
(14)$$\begin{array}{l} \;\frac{P_{2,k}\left(i,j+1\right)-2P_{2,k}\left(i,j\right)+P_{2,k}\left(i,j-1\right)}{∆t^{2}}+\gamma_{2}\frac{P_{2,k}\left(i,j\right)-P_{2,k}\left(i,j-1\right)}{∆t}\\ \;+4\pi^{2}f_{2}{}^{2}\left(P_{2,k}\left(i,j\right)\right)-\frac{4\pi^{2}f_{2}{}^{2}}{Q_{2}ed}{\left(P_{2,k}\left(i,j\right)\right)}^{2}+\frac{4\pi^{2}f_{2}{}^{2}}{Q_{2}{}^{2}e^{2}d^{2}}{\left(P_{2,k}\left(i,j\right)\right)}^{3}\\ \;=\frac{Q_{2}e^{2}}{m}\left(E_{k}\left(i,j\right)\right)\; \end{array}$$
(15)$$\begin{array}{l} \;\frac{P_{N,k}\left(i,j+1\right)-2P_{N,k}\left(i,j\right)+P_{N,k}\left(i,j-1\right)}{∆t^{2}}+\gamma_{N}\frac{P_{N,k}\left(i,j\right)-P_{N,k}\left(i,j-1\right)}{∆t}\\ \;+4\pi^{2}f_{N}{}^{2}\left(P_{N,k}\left(i,j\right)\right)-\frac{4\pi^{2}f_{N}{}^{2}}{Q_{N}ed}{\left(P_{N,k}\left(i,j\right)\right)}^{2}+\frac{4\pi^{2}f_{N}{}^{2}}{Q_{N}{}^{2}e^{2}d^{2}}{\left(P_{N,k}\left(i,j\right)\right)}^{3}\\ \;=\frac{Q_{N}e^{2}}{m}\left(E_{k}\left(i,j\right)\right) \end{array}$$
(16)$$P_{k}=P_{1,k}+P_{2,k}+\cdots +P_{N,k},\;Q=Q_{1}+Q_{2}+\cdots +Q_{N}$$

Equations (12)–(15) were solved for each new update. For high precision, we chose ∆t and ∆x to be small.

## 3. Non-Linear Programming for Efficient Harmonic Generation

As already mentioned, we used the BFGS algorithm for its relatively low computational cost and high convergence rate. The BFGS algorithm is a Quasi-Newton algorithm that recursively computes the Hessian matrix instead of computing the second derivative of the cost function at every iteration. The BFGS algorithm is useful when the Hessian matrix is not available or when it is too costly to compute.

### 3.1. BFGS Algorithm-Based Optimization

Start with an initial estimate of the Hessian matrix
(17)$$H_{0}=I\left(I:\mathrm{Identity}\;\mathrm{matrix}\right).$$

Identify the cost function that includes the penalty terms for constraint violations
(18)$$F\left(\nu \right)=\left|E\left(\nu =\nu_{target}\right)\right|=$$
$$\left|\int_{\nu_{target}-∆\nu }^{\nu_{target}+∆\nu }\left\{\int_{0}^{∆T}\{E\left(x=x^{\prime },t\right)e^{-i\left(2\pi \Omega \right)t}\}dt\right\}e^{i\left(2\pi \Omega \right)t}d\Omega \right|-L_{1}{\left\{\overset{N}{\underset{i=1}{\sum }}\delta_{i}\left(v_{i}-v_{max}\right)\right\}}^{q}-L_{2}{\left\{\overset{N}{\underset{i=1}{\sum }}\zeta_{i}\left(v_{min}-v_{i}\right)\right\}}^{q}$$
$$\delta_{i}=\left\{\begin{array}{c} 0\;\;\;\;\;\;\;\;\;\;\;\;\;if\;v_{i}\leq v_{max}\\ >0\;\;\;\;\;\;\;\;if\;v_{i}>v_{max} \end{array}\right\},\;\zeta_{i}=\left\{\begin{array}{c} 0\;\;\;\;\;\;\;\;\;\;\;\;\;if\;v_{i}\geq v_{min}\\ >0\;\;\;\;\;\;\;\;if\;v_{i}<v_{min} \end{array}\right\}$$
*q*: exponent of the penalty (positive valued), $\left\{L_{1},L_{2}\right\}\;$: penalty constants (positive valued), $\left\{\delta_{i},\zeta_{i}\right\}\;$: penalty weights

### 3.2. Determining the Penalty Weights

In order to decrease $\left|E\left(\nu =\nu_{target}\right)\right|$ by a factor of $\left(100-\frac{100}{\sigma }\right)\%$ for a deviation of ∆ν from $\nu_{\max}$ or $\nu_{\min}$, the penalty coefficients $\delta_{1}$ and $\delta_{2}$ were chosen as (assuming single-pulse tuning)
(19)$$\delta_{1}=\left\{\begin{array}{ll} 0 & if\;\;\;\nu_{p}<\nu_{max}\\ \frac{\left(1-\frac{1}{\sigma }\right)}{{\left(∆\nu \right)}^{2}}\left|E\left(\nu_{p}\right)\right| & if\;\;\;\nu_{p}>\nu_{max} \end{array}\right\},\;\delta_{2}=\left\{\begin{array}{ll} 0 & if\;\;\;\nu_{p}>\nu_{min}\\ \frac{\left(1-\frac{1}{\sigma }\right)}{{\left(∆\nu \right)}^{2}}\left|E\left(\nu_{p}\right)\right| & if\;\;\;\nu_{p}<\nu_{min} \end{array}\right\}$$

∆ν: deviation from max/min allowable frequency, σ : reduction factor.
(20)$$\mathrm{Such}\;\mathrm{that}\;F=\left|E\left(\nu =\nu_{target}\right)\right|-\delta_{1}{\left(\nu -\nu_{max}\right)}^{2}-\delta_{2}{\left(\nu_{min}-\nu \right)}^{2}$$

Identify the search direction, $p_{k}=-$
$H_{k}$
$\nabla F_{k}$ (Search direction). Determine a suitable step size (Backtracking approach). Select α>0,ρ ϵ (0,1),c ϵ (0,1). while $F(x_{k}+$
$\alpha \;p_{k})\leq$
$F\left(x_{k}\right)+$
$c\alpha \nabla F_{k}{}^{T}p_{k}\;\alpha \leftarrow \rho \alpha$ end.

Update the values of the optimization parameters: $\nu_{k+1}=\nu_{k}+\alpha_{k}$
$p_{k}$ (Update equation).

Compute the parameter difference vector and the gradient difference vector $s_{k}=\nu_{k+1}-$
$\nu_{k}$, $y_{k}=$
$\nabla F_{k+1}-$
$\nabla F_{k}$. Update the Hessian matrix for the next iteration $H_{k+1}=\left(I-\rho_{k}s_{k}y_{k}{}^{T}\right)H_{k}\left(I-\rho_{k}y_{k}s_{k}{}^{T}\right)+\rho_{k}s_{k}s_{k}{}^{T}$ (*BFGS update*)
(21)$$\nabla F=\left[\begin{array}{c} \frac{F\left(\nu_{1}+\epsilon ,\nu_{2},\ldots ,\nu_{N}\right)-F\left(\nu_{1},\nu_{2},\ldots ,\nu_{N}\right)}{\epsilon }\\ \frac{F\left(\nu_{1},\nu_{2}+\epsilon ,\ldots ,\nu_{N}\right)-F\left(\nu_{1},\nu_{2},\ldots ,\nu_{N}\right)}{\epsilon }\\ .\\ .\\ .\\ \frac{F\left(\nu_{1},\nu_{2},\ldots ,\nu_{N}+\epsilon \right)-F\left(\nu_{1},\nu_{2},\ldots ,\nu_{N}\right)}{\epsilon } \end{array}\right]\;,\;\rho_{k}=\frac{1}{y_{k}{}^{T}s_{k}}$$

Note that we solved the FDTD equations at every iteration of the BFGS algorithm until the desired intensity at a target frequency was obtained. The summary of the whole process is summarized in the flowchart presented in Figure 2.

## 4. Numerical Simulations

### 4.1. Simulation1: Intense Harmonic Generation in the Ultraviolet (UV) Frequency Range

The total excitation (pump) wave E that was composed of two high-intensity ultrashort pulses, energized a Fabry–Perot type optical micro-resonator that had an optical isolator at the input (left) port and a band-pass filter at the output (right) port. These two ultrashort pulses were both initiated at x = 2.5 μm at time t = 0 sec (Figure 3). The total excitation wave at the input port can be expressed as
(22)$$\begin{array}{c} E\left(x=0\;\mu m,t\right)=\overset{2}{\underset{i=1}{\sum }}A_{i}\cos\left(2\pi \nu_{i}t+\psi_{i}\right)(u\left(t\right)\\ \;-u\left(t-\Delta T_{i}\right))\frac{V}{m},\;u\left(t\right):Unit\;step\;function \end{array}$$
$$A_{1}=8\times 10^{7},\;A_{2}=1\times 10^{8},\;\Delta T_{1}=1.5\;\mathrm{ps},\;\Delta T_{2}=1\;\mathrm{ps}$$

Our goal was to generate a desired monochromatic wave (harmonic) at f = 820 THz, a typical UV-ablation frequency. To achieve this, the excitation frequencies of the ultrashort pulses were tuned.Spatial and temporal intervals of the simulation: 0≤x≤10 μm, 0≤t≤10 ps$\mathrm{Resonance}\;\mathrm{frequencies}\;\mathrm{of}\;\mathrm{the}\;\mathrm{interaction}\;\mathrm{medium}:\;f_{r}=\left\{4\times 10^{14}\;\mathrm{Hz},\;6.3\times 10^{14}\;\mathrm{Hz},\;8.8\times 10^{14}\;\mathrm{Hz}\right\}$$\mathrm{Polarization}\;\mathrm{damping}\;\mathrm{rates}\;\mathrm{of}\;\mathrm{the}\;\mathrm{interaction}\;\mathrm{medium}:\;\gamma =\left\{2\times 10^{9}\;\mathrm{Hz},\;3\times 10^{10}\;\mathrm{Hz},\;1\times 10^{11}\;\mathrm{Hz}\right\}$$\mathrm{Relative}\;\mathrm{permittivity}\;\mathrm{of}\;\mathrm{the}\;\mathrm{interaction}\;\mathrm{medium}:\;\left(\varepsilon_{r}\right)=10\;(\mu_{r}=1)$Location of the optical isolator: x=0 μm, Bandpass filter location: x=10 μmSpatial range of the interaction material :0 μm<x<10 μm$\mathrm{Density}\;\mathrm{of}\;\mathrm{electrons}:\;Q=3.5\times \frac{10^{28}}{m^{3}},\;\mathrm{Atom}\;\mathrm{diameter}:\;d=0.3\;\mathrm{nanometers}\;$Resonance probabilities: ξ={0.3,0.4,0.3}, Cost function to be maximized: C 
Problem statement: Identifying the excitation frequencies of the high-intensity ultrashort pulses for maximizing the intensity spectral density around the target frequency ($\left|E\left(\nu =\nu_{target}=820\mathrm{THz}\right)\right|$) inside the micro-resonator, for 50 THz < $\left\{\nu_{1},\nu_{2}\right\}<500\;\mathrm{THz}\;$, and for 0 μm<x<10 μm, 0≤t≤10 ps.
(23)$$\begin{array}{c} C=\left|E\left(\nu =\nu_{target}=820\mathrm{THz}\right)\right|\\ \;=\left|\int_{8.2\times 10^{14}-∆\nu }^{8.2\times 10^{14}+∆\nu }\left\{\int_{0}^{∆T}\{E_{in}\left(x=x^{\prime },t\right)e^{-i\left(2\pi \Omega \right)t}\}dt\right\}e^{i\left(2\pi \Omega \right)t}d\Omega \right| \end{array}$$


$$∆T=10\;\mathrm{ps},\;\left(8.2\times 10^{14}-∆\nu \right)\;\mathrm{Hz}<\Omega <\left(8.2\times 10^{14}+∆\nu \right)\;\mathrm{Hz},\;∆\nu =10\;\mathrm{THz}$$


Initial conditions of the electric field and polarization density: (Prime sign refers to the time derivative)
(24)$$\begin{array}{c} P_{2}\left(x,0\right)=P_{2}{}^{’}\left(x,0\right)=E_{2}\left(x,0\right)=E_{2}{}^{’}\left(x,0\right)=P_{1}\left(x,0\right)=P_{1}{}^{’}\left(x,0\right)=E_{1}\left(x,0\right)\\ =E_{1}{}^{’}\left(x,0\right)=0 \end{array}$$
**Bandpass filtering**: Frequency selective cavity wall (right port) is fixed at x=10 μm 
(25)$$\begin{array}{l} \left|\Gamma \left(\nu \right)\right|\\ =1-e^{-{\left(\frac{\left(\nu -820\mathrm{THz}\right)}{\sqrt{200\mathrm{THz}}}\right)}^{2}}\;\left(\mathrm{Magnitude}\;\mathrm{response}\;\mathrm{of}\;\mathrm{the}\;\mathrm{harmonic}\;\mathrm{selective}\;\mathrm{output}\;\mathrm{wall}\right) \end{array}$$
**Cost function:**
$C\left(\nu_{1},\nu_{2}\right)=\left|E\left(\nu =820\;\mathrm{THz}\right)\right|-\sum_{i=1}^{2}\{\delta_{i,1}{\left(\nu_{i}-\nu_{\max}\right)}^{2}+\delta_{i,2}{\left(\nu_{\min}-\nu_{i}\right)}^{2}\}\;$
(26)$$\begin{array}{c} \delta_{i,1}=\{\begin{array}{ll} \;0 & if\;\nu_{i}\leq \nu_{\max}\\ \frac{\left|E\left(\nu =820\;\mathrm{THz}\right)\right|}{10^{27}} & if\;\nu_{i}>\nu_{\max} \end{array}\},\\ \delta_{i,2}=\left\{\begin{array}{ll} \;0 & if\;\nu_{i}\geq \nu_{\min}\\ \frac{\left|E\left(\nu =820\;\mathrm{THz}\right)\right|}{10^{27}} & if\;\nu_{i}<\nu_{\min} \end{array}\right\} \end{array}$$
**FDTD Equations:** Discretization of Equations (1)–(4) via finite difference time domain is as follows [27,28,29,30]
(27)$$\begin{array}{l} \frac{E_{k}\left(i+1,j\right)-2E_{k}\left(i,j\right)+E_{k}\left(i-1,j\right)}{∆x^{2}}-{µ}_{0}\varepsilon_{\infty }\left(i,j\right)\frac{E_{k}\left(i,j+1\right)-2E_{k}\left(i,j\right)+E_{k}\left(i,j-1\right)}{∆t^{2}}\\ \;={µ}_{0}\sigma \left(i,j\right)\frac{E_{k}\left(i,j\right)-E_{k}\left(i,j-1\right)}{∆t}\\ \;+{µ}_{0}\frac{P_{k}\left(i,j+1\right)-2P_{k}\left(i,j\right)+P_{k}\left(i,j-1\right)}{\Delta t^{2}} \end{array}$$
(28)$$\begin{array}{l} \frac{P_{1,k}\left(i,j+1\right)-2P_{1,k}\left(i,j\right)+P_{1,k}\left(i,j-1\right)}{∆t^{2}}+\gamma_{1}\frac{P_{1,k}\left(i,j\right)-P_{1,k}\left(i,j-1\right)}{∆t}\\ \;+4\pi^{2}f_{1}{}^{2}\left(P_{1,k}\left(i,j\right)\right)-\frac{4\pi^{2}f_{1}{}^{2}}{Q_{1}ed}{\left(P_{1,k}\left(i,j\right)\right)}^{2}+\frac{4\pi^{2}f_{1}{}^{2}}{Q_{1}{}^{2}e^{2}d^{2}}{\left(P_{1,k}\left(i,j\right)\right)}^{3}\\ \;=\frac{Q_{1}e^{2}}{m}\left(E_{k}\left(i,j\right)\right). \end{array}$$
(29)$$\begin{array}{l} \frac{P_{2,k}\left(i,j+1\right)-2P_{2,k}\left(i,j\right)+P_{2,k}\left(i,j-1\right)}{∆t^{2}}+\gamma_{2}\frac{P_{2,k}\left(i,j\right)-P_{2,k}\left(i,j-1\right)}{∆t}\\ \;+4\pi^{2}f_{2}{}^{2}\left(P_{2,k}\left(i,j\right)\right)-\frac{4\pi^{2}f_{2}{}^{2}}{Q_{2}ed}{\left(P_{2,k}\left(i,j\right)\right)}^{2}+\frac{4\pi^{2}f_{2}{}^{2}}{Q_{2}{}^{2}e^{2}d^{2}}{\left(P_{2,k}\left(i,j\right)\right)}^{3}\\ \;=\frac{Q_{2}e^{2}}{m}\left(E_{k}\left(i,j\right)\right). \end{array}$$
(30)$$\begin{array}{l} \frac{P_{3,k}\left(i,j+1\right)-2P_{3,k}\left(i,j\right)+P_{3,k}\left(i,j-1\right)}{∆t^{2}}+\gamma_{3}\frac{P_{3,k}\left(i,j\right)-P_{3,k}\left(i,j-1\right)}{∆t}\\ \;+4\pi^{2}f_{3}{}^{2}\left(P_{3,k}\left(i,j\right)\right)-\frac{4\pi^{2}f_{3}{}^{2}}{Q_{3}ed}{\left(P_{3,k}\left(i,j\right)\right)}^{2}+\frac{4\pi^{2}f_{3}{}^{2}}{Q_{3}{}^{2}e^{2}d^{2}}{\left(P_{3,k}\left(i,j\right)\right)}^{3}\\ \;=\frac{Q_{3}e^{2}}{m}\left(E_{k}\left(i,j\right)\right). \end{array}$$
P=$P_{1}+P_{2}+P_{3}$*x*: Space coordinate, *t*: Time, *k*: Iteration, $E_{k}\left(x,t\right)=E_{k}\left(i∆x,j∆t\right)\;\to E_{k}\left(i,j\right)$
$$E_{k}:\;\mathrm{Total}\;\mathrm{wave}\;\mathrm{electric}\;\mathrm{field}\;\mathrm{at}\;\mathrm{the}\;\mathrm{kth}\;\mathrm{update}$$
**Optimization via BFGS algorithm:** Choose the identity matrix as the initial Hessian matrix
$$H_{0}=I\;\left(I:2\times 2\;\mathrm{identity}\;\mathrm{matrix}\right)$$
$$v_{p1,0}=250\;\mathrm{THz},\;\;v_{p1,1}=245\;\mathrm{THz},\;\;v_{p2,0}=225\;\mathrm{THz},\;\;v_{p2,1}=220\;\mathrm{THz},\;\alpha_{1}=0.4$$
(31)$$\nabla C_{k}=\left[\begin{array}{c} \frac{C\left(f_{p1,k},\;f_{p2,k}\right)-C\left(f_{p1,k-1},\;f_{p2,k}\right)}{f_{p1,k}-f_{p1,k-1}}\\ \frac{C\left(f_{p1,k},\;f_{p2,k}\right)-C\left(f_{p1,k},\;f_{p2,k-1}\right)}{f_{p2,k}-f_{p2,k-1}} \end{array}\right]$$
$p_{k}=-$$H_{k}$$\nabla C_{k}$, $\nu_{\;k+1}=\nu_{k}+\alpha_{k}$$p_{k}$, $s_{k}=v_{k+1}-$$v_{k},$$\nu_{k}=\left[\begin{array}{c} \nu_{1,k}\\ \nu_{2,k} \end{array}\right]$$y_{k}=$$\nabla C_{k+1}-$$\nabla C_{k}$$\rho_{k}=\frac{1}{y_{k}{}^{T}s_{k}}$
BFGS recursion: $H_{k+1}=\left(I-\rho_{k}s_{k}y_{k}{}^{T}\right)H_{k}\left(I-\rho_{k}y_{k}s_{k}{}^{T}\right)+\rho_{k}s_{k}s_{k}{}^{T}$I:Identity matrix
(32)$$\nabla C_{k+1}=\left[\begin{array}{c} \frac{C\left(f_{p1,k+1},\;f_{p2,k}\right)-C\left(f_{p1,k},\;f_{p2,k}\right)}{f_{p1,k+1}-f_{p1,k}}\\ \frac{C\left(f_{p1,k},\;f_{p2,k+1}\right)-C\left(f_{p1,k},\;f_{p2,k}\right)}{f_{p2,k+1}-f_{p2,k}} \end{array}\right]$$
A simple formula for the step size that complies with the backtracking approach is determined as
(33)$$\alpha_{k}=c^{(\;\log\left|\frac{C\left(\nu_{k}\right)}{C\left(\nu_{k}\right)-C\left(\nu_{k-1}\right)}\right|\;)\;/\;\left(\;\left|\frac{C\left(\nu_{k}\right)}{C\left(\nu_{k}\right)-C\left(\nu_{k-1}\right)}\right|\;\right)}$$

The base of the step size (c) was a simple constant (1<c<1.5 ) and $\alpha_{k}$ was the step size at iteration k. In this example, c was chosen as c = 1.45. According to these presented formulations, the maximum amplitude of the desired harmonic that was attained in the micro-resonator (for 0 < t < 10 ps) was obtained as $\left|E\left(\nu =\nu_{target}=820\;\mathrm{THz}\right)\right|=7.6\times 10^{7}\;V/m\;$, and the resulting optimal excitation frequencies were identified as $\nu_{1}=273.2\;\mathrm{THz},\;\nu_{2}=284.7\;\mathrm{THz}\;\;$(see Table 1).
(34)$$\begin{array}{l} W_{e,p}=\mathrm{Stored}\;\mathrm{electric}\;\mathrm{energy}\;\mathrm{density}=\frac{1}{2}\varepsilon_{\infty }E^{2}+\frac{1}{2}EP\;\left(\frac{Joules}{m^{3}}\right),\\ E\;:\mathrm{Electric}\;\mathrm{field}\;\mathrm{intensity} \end{array}$$
P: Polarization density created by the pump wave
$\left(\frac{Coulomb}{m^{2}}\right),\;\varepsilon_{\infty }:\mathrm{Background}\;\mathrm{permittivity}.$

As indicated in Table 1, the optimal frequencies of the ultrashort excitation pulses gave rise to an enormous amount of stored electric energy and a corresponding large amount of stored polarization density (non-linear coupling coefficient). When we investigated Table 1, we realized that the induced electric energy density and the induced polarization density should be concurrently high for a super-intense harmonic generation at a desired frequency. The amount of stored electric energy was crucial in generating the high-intensity target harmonic and the induced polarization density was important in concentrating the available spectral energy around the desired frequency and in transferring the energy from one harmonic to another. It is important to note that in a multi-resonant interaction medium with many resonances, if the desired frequency (target harmonic) is near one of the resonance frequencies of the medium, generating the target harmonic with a high intensity is harder as the dielectric absorption is stronger near the resonance frequencies of the medium. Therefore, if the desired frequency of the harmonic is near any resonance, one should expect a relatively lower-intensity target harmonic after the optimization process is completed. To provide an increase in the intensity of the desired harmonic, one may use more ultrashort excitation pulses. At the end of 13th iteration, electric field at the output of the bandpass filter for 10 ps simulation time is seen in Figure 4. As seen in this figure, electric field at 820 THz reaches in a scale of 10^8^ V/m.

In Figure 5 we present spectral magnitude through the iterations. As seen in Figure 5a there is no generated target frequency (820 THz) at 6th iteration. When the iterations proceed (Figure 5b–d), wave mixing between two source waves happens, and we obtained generated target frequency at 820 THz. At the end of 13th iteration, intensity spectral density at the band-pass filter output for 822 THz is 1023 (W/m^2^Hz) as seen in Figure 6.

### 4.2. Simulation 2: Intense Quasi-Monochromatic Yellow-Light Generation Around 515THz

The total excitation (pump) wave E that was composed of two high-intensity ultrashort pulses, energized a Fabry–Perot type optical micro-resonator that had an optical isolator at the input (left) port and a band-pass filter at the output (right) port as seen in Figure 7. All ultrashort pulses were initiated at *x* = 2.5 μm at time t = 0 sec. The total excitation wave at the input port can be expressed as
(35)$$E\left(x=0\;\mu m,t\right)=\overset{2}{\underset{i=1}{\sum }}A_{i}\cos\left(2\pi \nu_{i}t+\psi_{i}\right)\left(u\left(t\right)-u\left(t-\Delta T_{i}\right)\right)\;V/m$$
$$A_{1}=1\times 10^{8},\;A_{2}=1.2\times 10^{8},\;\Delta T_{1}=1\mathrm{ps},\;\Delta T_{2}=0.7\mathrm{ps}$$
Our goal was to generate a desired monochromatic wave (harmonic) at f = 515 THz. To achieve this, the excitation frequencies of the ultrashort pulses were tuned.Spatial and temporal intervals of the simulation: 0≤x≤10 μm, 0≤t≤10ps$\mathrm{Resonance}\;\mathrm{frequencies}\;\mathrm{of}\;\mathrm{the}\;\mathrm{interaction}\;\mathrm{medium}:\;f_{r}=\left\{3\times 10^{14}\mathrm{Hz},\;4.4\times 10^{14}\mathrm{Hz},\;6.3\times 10^{14}\mathrm{Hz}\right\}$$\mathrm{Polarization}\;\mathrm{damping}\;\mathrm{rates}\;\mathrm{of}\;\mathrm{the}\;\mathrm{interaction}\;\mathrm{medium}:\;\gamma =\left\{1\times 10^{10}\mathrm{Hz},\;2.5\times 10^{10}\mathrm{Hz},\;1\times 10^{11}\mathrm{Hz}\right\}$Resonance probabilities: $\xi =\left\{\frac{1}{3},\frac{1}{3},\frac{1}{3}\right\}\;$, $\mathrm{Permittivity}\;\mathrm{of}\;\mathrm{the}\;\mathrm{interaction}\;\mathrm{medium}:\;\left(\varepsilon_{r}\right)=12\;(\mu_{r}=1)\;$Location of the optical isolator (input port): x=0 μm, Filter (output port) location: x=10 μmSpatial range of the interaction material :0 μm<x<10 μm, $\mathrm{Density}\;\mathrm{of}\;\mathrm{electrons}:\;N=3.5\times \frac{10^{28}}{m^{3}}\;$Atom diameter: d=0.3 nanometers, Cost function to be maximized: C
Problem statement: Identifying the excitation frequencies of the high-intensity ultrashort pulses for maximizing the intensity spectral density around a target frequency ($\left|E\left(\nu =\nu_{target}=515\;\mathrm{THz}\right)\right|$) inside the micro-resonator, for 50THz < $\left\{\nu_{1},\nu_{2}\right\}<500\;\mathrm{THz}\;$, and for 0 μm<x<10 μm, 0≤t≤10 ps.
(36)$$\begin{array}{c} C=\left|E\left(\nu =\nu_{target}=515THz\right)\right|\\ =\left|\int_{5.15\times 10^{14}-∆\nu }^{5.15\times 10^{14}+∆\nu }\left\{\int_{0}^{∆T}\{E_{\mathrm{in}}\left(x=x^{\prime },t\right)e^{-i\left(2\pi \Omega \right)t}\}dt\right\}e^{i\left(2\pi \Omega \right)t}d\Omega \right| \end{array}$$

$$∆T=10\mathrm{ps},\;\left(5.15\times 10^{14}-∆\nu \right)\mathrm{Hz}<\Omega <\left(5.15\times 10^{14}+∆\nu \right)\mathrm{Hz},\;∆\nu =10\mathrm{THz}$$**Initial conditions of the electric field and polarization density**: (Prime sign refers to the time derivative)
$$P_{2}\left(x,0\right)=P_{2}{}^{\prime }\left(x,0\right)=E_{2}\left(x,0\right)=E_{2}{}^{\prime }\left(x,0\right)=P_{1}\left(x,0\right)=P_{1}{}^{\prime }\left(x,0\right)=E_{1}\left(x,0\right)=E_{1}{}^{\prime }\left(x,0\right)=0$$**Band-pass filtering**: Frequency selective cavity wall (right port) is fixed at x=10μm (37)$$\left|\Gamma \left(\nu \right)\right|=1-e^{-{\left(\frac{\left(\nu -515\mathrm{THz}\right)}{\sqrt{200}\mathrm{THz}}\right)}^{2}}$$**Cost function:**$C\left(\nu_{1},\nu_{2}\right)=\left|E\left(\nu =515\mathrm{THz}\right)\right|-\sum_{i=1}^{2}\{\delta_{i,1}{\left(\nu_{i}-\nu_{max}\right)}^{2}+\delta_{i,2}{\left(\nu_{min}-\nu_{i}\right)}^{2}\}\;$(38)$$\begin{array}{c} \delta_{i,1}=\{\begin{array}{ll} \;0 & if\;\nu_{i}\leq \nu_{max}\\ \frac{\left|E\left(\nu =515\mathrm{THz}\right)\right|}{10^{27}} & \;if\;\nu_{i}>\nu_{max} \end{array}\},\\ \delta_{i,2}=\left\{\begin{array}{ll} \;0 & if\;\nu_{i}\geq \nu_{min}\\ \frac{\left|E\left(\nu =515\mathrm{THz}\right)\right|}{10^{27}} & \;if\;\nu_{i}<\nu_{min} \end{array}\right\} \end{array}$$**Optimization via BFGS algorithm:** Choose the identity matrix as the initial Hessian matrix
$$H_{0}=I\;\left(I:2\times 2\;\mathrm{identity}\;\mathrm{matrix}\right)$$
$$f_{p1,0}=250\;\mathrm{THz},\;\;f_{p1,1}=245\;\mathrm{THz},\;\;f_{p2,0}=225\;\mathrm{THz},\;\;f_{p2,1}=220\;\mathrm{THz},\;\alpha_{1}=0.5$$
(39)$$\nabla C_{k}=\left[\begin{array}{c} \frac{C\left(f_{p1,k},\;f_{p2,k}\right)-C\left(f_{p1,k-1},\;f_{p2,k}\right)}{f_{p1,k}-f_{p1,k-1}}\\ \frac{C\left(f_{p1,k},\;f_{p2,k}\right)-C\left(f_{p1,k},\;f_{p2,k-1}\right)}{f_{p2,k}-f_{p2,k-1}} \end{array}\right],\;\nabla C_{k+1}=\left[\begin{array}{c} \frac{C\left(f_{p1,k+1},\;f_{p2,k}\right)-C\left(f_{p1,k},\;f_{p2,k}\right)}{f_{p1,k+1}-f_{p1,k}}\\ \frac{C\left(f_{p1,k},\;f_{p2,k+1}\right)-C\left(f_{p1,k},\;f_{p2,k}\right)}{f_{p2,k+1}-f_{p2,k}} \end{array}\right]$$
$p_{k}=-$$H_{k}$$\nabla C_{k}$, $\nu_{\;k+1}=\nu_{k}+\alpha_{k}$$p_{k}$, $s_{k}=f_{p,k+1}-$$f_{p,k},$$\nu_{k}=\left[\begin{array}{c} \nu_{1,k}\\ \nu_{2,k} \end{array}\right]$, $y_{k}=$$\nabla C_{k+1}-$$\nabla C_{k}$, $\rho_{k}=\frac{1}{y_{k}{}^{T}s_{k}}$*BFGS recursion:*$H_{k+1}=\left(I-\rho_{k}s_{k}y_{k}{}^{T}\right)H_{k}\left(I-\rho_{k}y_{k}s_{k}{}^{T}\right)+\rho_{k}s_{k}s_{k}{}^{T}$I:Identity matrix
(40)$$\alpha_{k}=c^{(\;\log\left|\frac{C\left(\nu_{k}\right)}{C\left(\nu_{k}\right)-C\left(\nu_{k-1}\right)}\right|\;)\;/\;\left(\;\left|\frac{C\left(\nu_{k}\right)}{C\left(\nu_{k}\right)-C\left(\nu_{k-1}\right)}\right|\;\right)}$$

The base of the step size (c) was a simple constant (1<c<1.5 ) and $\alpha_{k}$ was the step size at iteration k. In this example, c was chosen as c = 1.40. According to these presented formulations, the maximum amplitude of the desired harmonic that was attained in the micro-resonator (for 0 < t < 10 ps) was obtained as $\left|E\left(\nu =\nu_{target}=515\;\mathrm{THz}\right)\right|=9.1\times 10^{7}\;V/m\;$, and the resulting optimal excitation frequencies were determined as $\nu_{1}=266.2\;\mathrm{THz},\;\nu_{2}=476.5\;\mathrm{THz}\;$(Table 2).
(41)$$\begin{array}{l} W_{e,p}=\mathrm{Stored}\;\mathrm{electric}\;\mathrm{energy}\;\mathrm{density}=\frac{1}{2}\varepsilon_{\infty }E^{2}+\frac{1}{2}EP\;\left(\frac{Joules}{m^{3}}\right),\;\\ E\;:\mathrm{Electric}\;\mathrm{field}\;\mathrm{intensity} \end{array}$$
P: Polarization density created by the pump wave
$\left(\frac{Coulomb}{m^{2}}\right),\;\varepsilon_{\infty }:\mathrm{Background}\;\mathrm{permittivity}.$

At the end of 33th iteration, electric field at the output of the bandpass filter for 10 ps simulation time is seen in Figure 8. As seen in this figure, electric field at 515 THz reaches in a scale of 10^8^ V/m.

In Figure 9 we present spectral magnitude through the iterations. As seen in Figure 9a there is no generated target frequency (515 THz) at 6th iteration. When the iterations proceed (Figure 9b–d), wave mixing between two source waves happens, and we obtained generated target frequency at 515 THz. Compared to generation of a wave at 820 THz we obtained much more sharp spectral distribution around the target frequency of 515 THz. At the end of 33th iteration, intensity spectral density at the band-pass filter output for 515 THz is 234.1 (W/m^2^Hz) as seen in Figure 10.

## 5. Testing the Model Accuracy via Comparison with Experimental Results

In order to validate the accuracy of our model, we compared our numerical results with the well-established experimental formula of second harmonic generation efficiency for many different excitation amplitudes. The following example illustrates this comparison by computing the error percentage for different excitation amplitudes. Once the error percentage was below one percent for every excitation amplitude, we assessed the computational results to be in good agreement with the experimental results.

### Example 5.1: Second Harmonic Generation by Nonlinear Wave Mixing

The goal of this example was to compute the second harmonic generation efficiency of a high intensity input wave after it had propagated through an interaction medium for a certain amount of time. The input wave was initially monochromatic with an angular frequency of $\omega_{1}$. The generated second harmonic of the input wave had an angular frequency of $\omega_{2}=2\omega_{1}$. Experimental and computational formulations were compared.

The 100 THz (monochromatic) high-amplitude input wave $E_{1}$ is excited at x = 2.4 μm (see Figure 11). The excitation amplitude of the input wave is $A_{1}$ (V/m).$E_{1}\left(x=2.4\;\mu m,t\right)=A_{1}\times \sin\left(2\pi \left(1\times 10^{14}\right)t+\varphi_{1}\right)\;V/m$$\left(\;\varphi_{1}=0\right)\;$Spatial range and duration of the computation: 0≤x≤10 μm, 0≤t≤30 ps $\mathrm{Resonance}\;\mathrm{frequencies}\;\mathrm{of}\;\mathrm{the}\;\mathrm{interaction}\;\mathrm{medium}:\;f_{r}=\left\{7.8\times 10^{14}\;\mathrm{Hz},\;9.5\times 10^{14}\;\mathrm{Hz},\;1.4\times 10^{15}\;\mathrm{Hz}\right\}$$\mathrm{Damping}\;\mathrm{coefficients}\;\mathrm{of}\;\mathrm{the}\;\mathrm{interaction}\;\mathrm{medium}:\;\gamma =\left\{4\times 10^{12}\;\mathrm{Hz},\;3\times 10^{12}\;\mathrm{Hz},\;1\times 10^{12}\;\mathrm{Hz}\right\}\;$$\mathrm{Interaction}\;\mathrm{medium}\;\mathrm{background}\;\mathrm{permitttivity}\;\left(\varepsilon_{\infty }\right)=1+\chi =12\;(\mu_{r}=1)$Left absorption domain ranges from x=0 to x=2.35 μm (absorbing boundary)Right absorption domain ranges from x=7.65 μm to x=10 μm (absorbing boundary) 
The experimental formula for the efficiency of second harmonic generation is stated as [40,41,42]
$$\eta_{\mathrm{experimental}}={\left(\tanh\sqrt{d^{2}\eta^{3}\omega^{2}cn\varepsilon_{0}A_{1}{}^{2}L^{2}}\;\right)}^{2}\;$$
*d* = Nonlinearity coefficient, η = Medium impedance, *n* = Index of refraction$A_{1}$ = Input wave amplitude, *L* = Length of the medium
$$\omega_{2}=\mathrm{Angular}\;\mathrm{frequency}\;\mathrm{of}\;\mathrm{the}\;\mathrm{generated}\;\sec\mathrm{ond}\;\mathrm{harmonic}$$

The computational results were obtained using Equations (12)–(16).

For a computation range of $0\leq t\leq t_{\max}\;$, the expression for the computational second harmonic generation efficiency is stated as [40,41,42]
(42)$$\eta_{\mathrm{computational}}=\frac{\mathrm{Intensity}\;\mathrm{of}\;\mathrm{the}\;\sec\mathrm{ond}\;\mathrm{harmonic}\;\mathrm{of}\;\mathrm{the}\;\mathrm{source}\;\mathrm{wave}\;\mathrm{at}\;t=t_{\max}}{\;\mathrm{Intensity}\;\mathrm{of}\;\mathrm{the}\;\mathrm{first}\;\mathrm{harmonic}\;\mathrm{of}\;\mathrm{the}\;\mathrm{source}\;\mathrm{wave}\;\mathrm{at}\;t=0\;}$$
The parameters of the computation are as follows:
$$\omega_{1}=\mathrm{First}\;\mathrm{harmonic}\;\mathrm{angular}\;\mathrm{frequency}=\left(2\pi \times 100\right)\;\mathrm{THz}$$
*L* = Length of the interaction medium = *3.33*
μm (ranging from *x* = 3.33 μm to 6.66 μm)$\omega_{2}=\mathrm{Second}\;\mathrm{harmonic}\;\mathrm{angular}\;\mathrm{frequency}=2\pi \times 200\;\mathrm{THz},$*n*$=\sqrt{12}$$A_{1}$ = Input wave amplitude (Varied from $10^{8}\frac{V}{m}to\;2.5\times 10^{9}\;V/m$, in increment of $10^{8}\;V/m$)

Oscillator weights (Resonance probabilities) = ξ = {0.3, 0.4, 0.3}

Based on Equation (42), the computational second harmonic efficiency $\left(\eta_{\mathrm{computational}}\right)$ was found as $\eta_{\mathrm{computational}}=1.74\times 10^{-4}$ for an excitation amplitude of $A_{1}=3\times 10^{8\;}V/m$. Using this result, we tried to estimate the nonlinear coefficient of the medium by solving the following transcendental equation [40,41,42]
(43)$${\left(\tanh\sqrt{d_{est}{}^{2}\eta^{3}\omega^{2}cn\varepsilon_{0}A_{1}{}^{2}L^{2}}\;\right)}^{2}=1.21\times \frac{10^{-21}C}{V}$$

From which we solved for the estimated nonlinear coefficient as $d_{est}=1.21\times 10^{-21}C/V$. However, in order to verify this estimated nonlinearity coefficient, we had to check whether this coefficient matched the theoretical results with the computational results for all input wave amplitudes. This verification is shown below in Table 3 for some sample excitation amplitudes and illustrated in Figure 12 for a broad range of excitation amplitudes. The theoretical and computational results seem to be in good agreement, there is more than 98% accuracy between the theoretical and the numerical results.

## 6. Conclusions

For an arbitrary micro-resonator interaction medium of multiple resonances, the frequencies of the ultrashort excitation pulses must be tuned accordingly, based on a rapidly converging and computationally efficient Quasi-Newton algorithm (such as BFGS) in order to generate a desired harmonic at an ultra-high efficiency. The non-linear programming process is basically needed to maximize the stored spectral energy density around the desired frequency so that the conversion efficiency is maximized. For a micro-resonator medium with many resonance frequencies, the computational cost increases as the number of differential equations increases due to the occurrence of many polarization density components. However, the cost can be reduced by decreasing the number of excitation pulses and alleviating the burden on non-linear programming. Even a single parameter optimization via a single excitation pulse may allow us to achieve the target harmonic generation efficiency. However, for devices that have very limited output powers, multiple excitation pulses must be adjusted in order to achieve the optimal frequency combination that allows the highest efficiency to be attained and to compensate for the output-power limitations of the source device.

## Figures and Tables

**Figure 1 micromachines-11-00686-f001:**
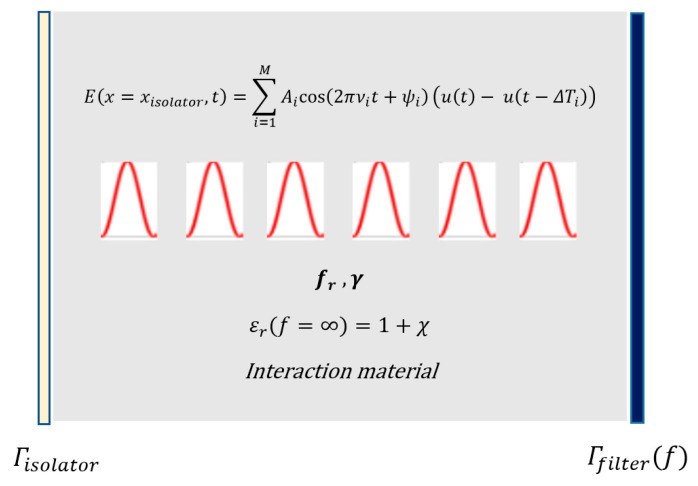
Excitation of an optical microcavity using M different ultrashort pulses, whose frequencies are to be tuned for high-intensity targeted harmonic generation via intracavity energy maximization.

**Figure 2 micromachines-11-00686-f002:**
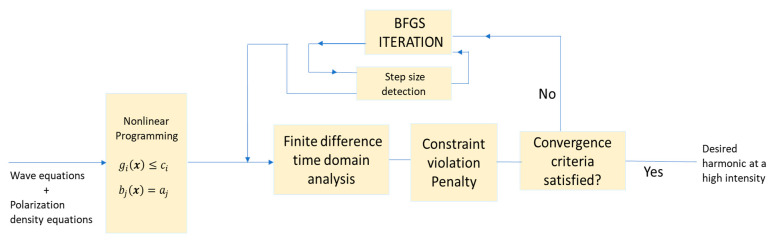
Flowchart description of high-intensity harmonic generation using non-linear programming.

**Figure 3 micromachines-11-00686-f003:**
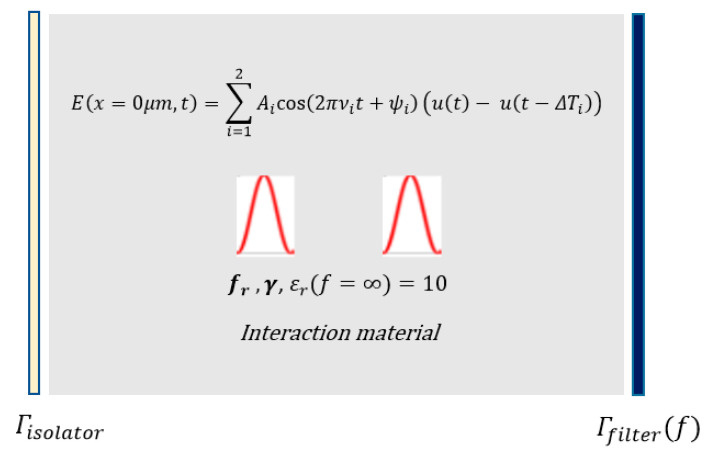
Nonlinear mixing of two ultrashort pulses in an optical microcavity, whose frequencies were tuned for high-intensity targeted harmonic generation at 820THz via intracavity energy maximization.

**Figure 4 micromachines-11-00686-f004:**
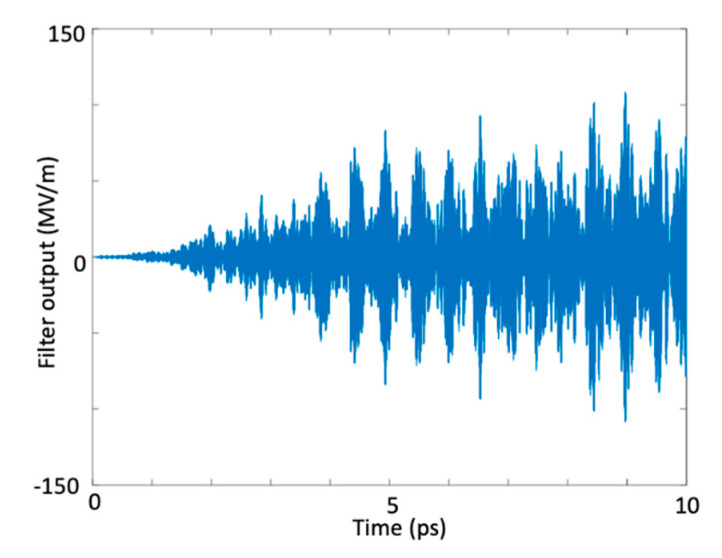
Time variation of the maximum electric field amplitude at the bandpass filter output (820 THz).

**Figure 5 micromachines-11-00686-f005:**
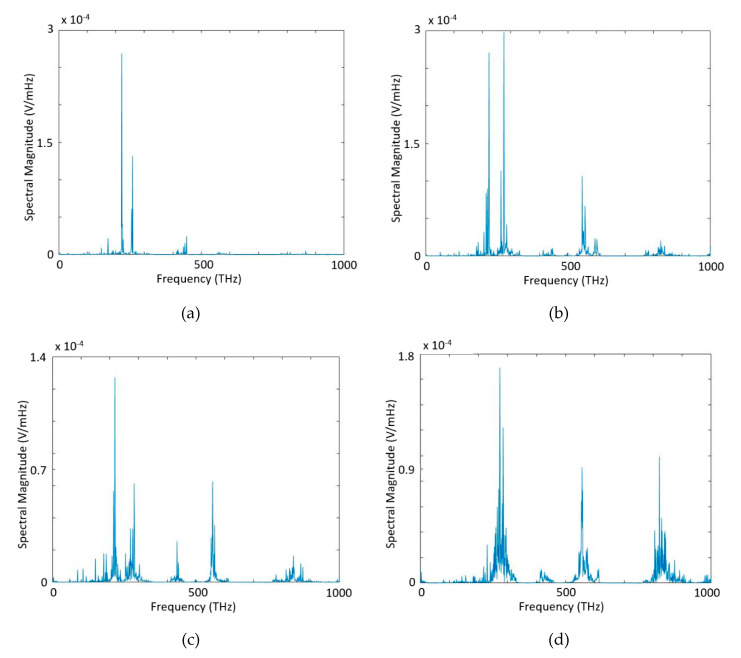
Spectrum of the total wave inside the cavity measured at x = 5.73 µm, t = 10 ps, at (**a**) 6th, (**b**) 8th, (**c**) 10th, and (**d**) 13th iteration of the optimization process for harmonic generation at 820 THz.

**Figure 6 micromachines-11-00686-f006:**
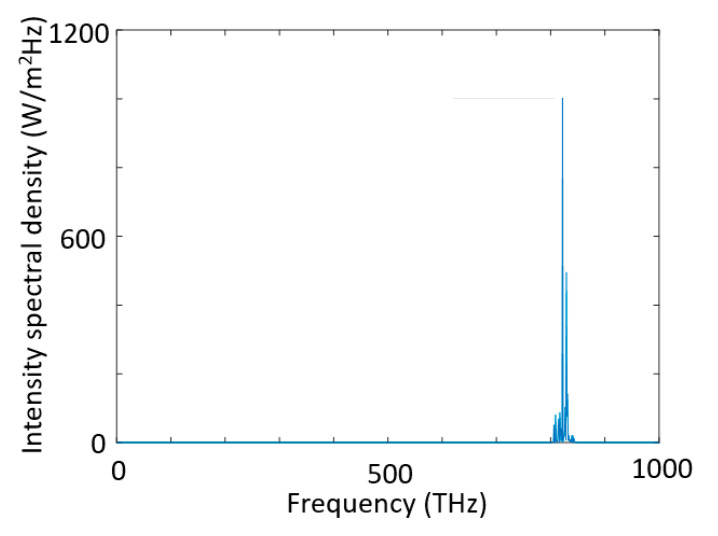
Intensity spectral density at the band-pass filter output for harmonic generation at 820 THz.

**Figure 7 micromachines-11-00686-f007:**
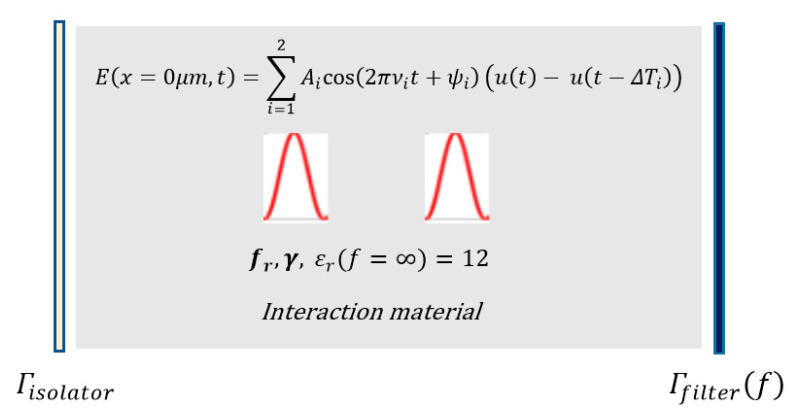
Nonlinear mixing of two ultrashort pulses in an optical microcavity, whose frequencies were tuned for high-intensity targeted harmonic generation at 515THz.

**Figure 8 micromachines-11-00686-f008:**
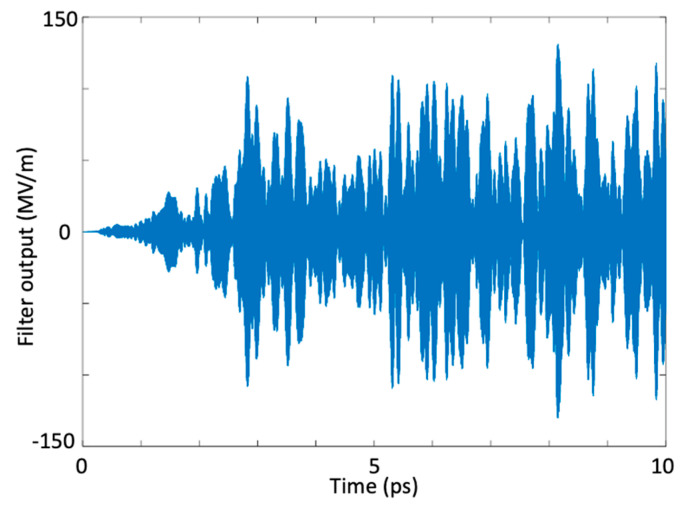
Time variation of the maximum amplitude at the band-pass filter output (harmonic generation at 515 THz).

**Figure 9 micromachines-11-00686-f009:**
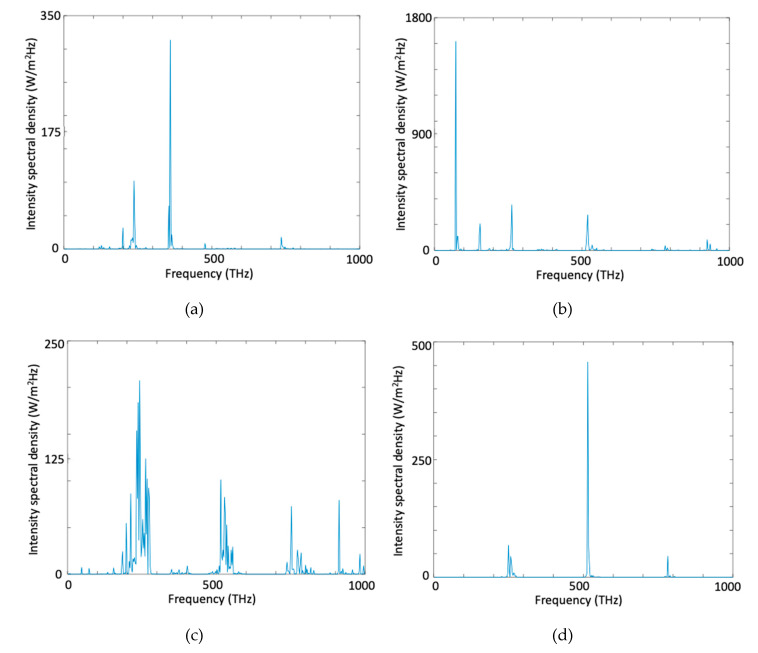
Spectrum of the total wave inside the cavity, measured at x = 5.73 µm, t = 10 ps, at (**a**) 6th, (**b**) 14th, (**c**) 24th, and (**d**) 33rd iteration of the optimization process for harmonic generation at 515 THz.

**Figure 10 micromachines-11-00686-f010:**
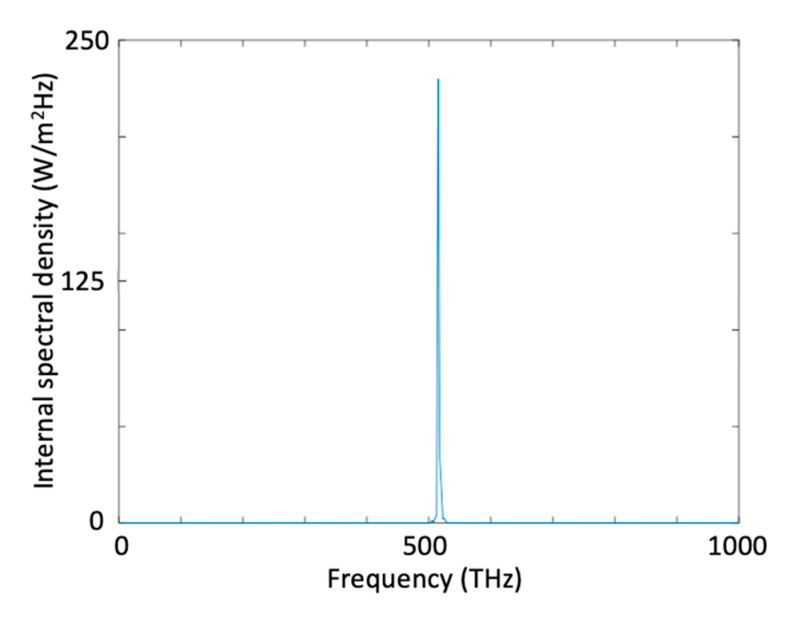
Intensity spectral density at the band-pass filter output for generation of a wave at 515 THz.

**Figure 11 micromachines-11-00686-f011:**
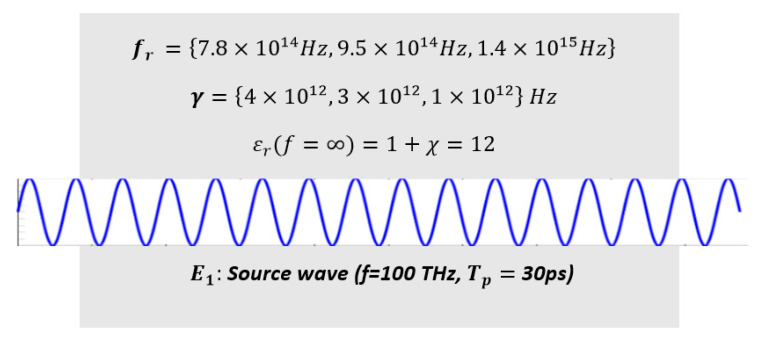
Configuration for Example 5.1.

**Figure 12 micromachines-11-00686-f012:**
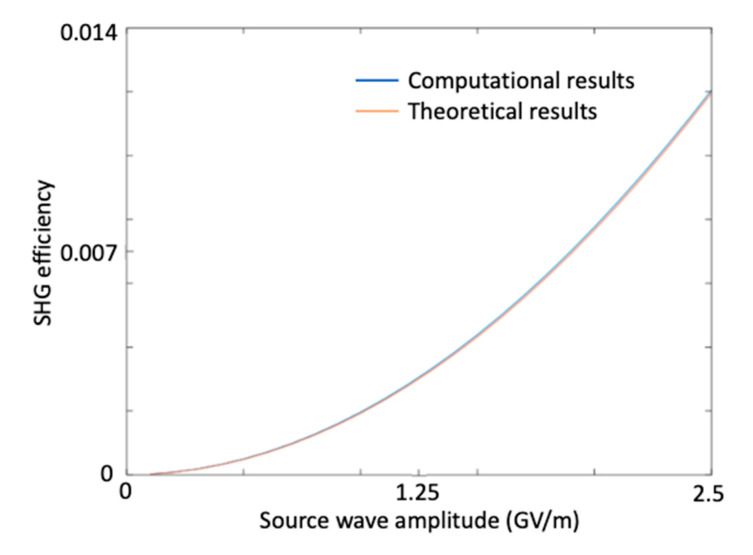
Comparison of the theoretical and the computational second harmonic generation efficiencies for $f_{\mathrm{input}}$ = 100 THz and d = $1.21\times 10^{-21}$ C/V, versus the source wave amplitude.

**Table 1 micromachines-11-00686-t001:** Broyden–Fletcher–Goldfarb–Shanno (BFGS) updating process.

$\left|E_{\nu =820\;\mathrm{THz}}\right|$	$\nu_{1}$	$\nu_{2}$	$W_{e,p}\;\left(\frac{J}{m^{3}}\right)$	$P_{pump}\left(\frac{C}{m^{2}}\right)$	*k* (Iteration #)
5.8 $\times 10^{3}\;V/m$	250 THz	225 THz	2.9 $\times 10^{7}$	0.09	1
2.9 $\times 10^{3}\;V/m$	245 THz	220 THz	1 $\times 10^{7}\;$	0.05	2
2.7 $\times 10^{5}\;V/m\;$	249.4 THz	226.1 THz	8.66 $\times 10^{7}\;$	0.14	3
2.0 $\times 10^{4}\;V/m\;$	257.2 THz	231.3 THz	5.49 $\times 10^{7}\;$	0.13	4
1.3 $\times 10^{3}\;V/m\;$	264.9 THz	226.6 THz	1.31 $\times 10^{7}\;$	0.06	5
7.6 $\times 10^{3}\;V/m\;$	276.0 THz	222.2 THz	1.76 $\times 10^{7}\;$	0.07	6
1.4 $\times 10^{4}\;V/m\;$	286.7 THz	231.8 THz	1.73 $\times 10^{7}\;$	0.07	7
1.1 $\times 10^{6}\;V/m\;$	279.7 THz	218.0 THz	3.06 $\times 10^{7}\;$	0.10	8
9.8 $\times 10^{5}\;V/m\;$	272.1 THz	236.9 THz	2.49 $\times 10^{7}\;$	0.10	9
5.1 $\times 10^{6}\;V/m\;$	263.5 THz	285.1 THz	3.30 $\times 10^{7}\;$	0.10	10
1.2 $\times 10^{7}\;V/m\;$	273.4 THz	310.7 THz	8.21 $\times 10^{7}\;$	0.16	11
3.2 $\times 10^{6}\;V/m\;$	275.1 THz	288.8 THz	3.48 $\times 10^{7}\;$	0.12	12
7.6 $\times 10^{7}\;V/m\;$	273.2 THz	284.7 THz	5.75 $\times 10^{7}\;$	0.17	13

**Table 2 micromachines-11-00686-t002:** BFGS updating process.

$\left|E_{\nu =515\mathrm{THz}}\right|$	$\;\nu_{1}\;$	$\;\nu_{2}\;$	$\;W_{e,p}\;\left(\frac{J}{m^{3}}\right)\;$	$P_{pump}\left(\frac{C}{m^{2}}\right)\;$	k (iteration #)
1.9 $\times 10^{3}\;V/m\;$	250 THz	225 THz	2.4 $\times 10^{7}$	0.20	1
2.6 $\times 10^{3}\;V/m\;$	245 THz	220 THz	$\;2.5\times 10^{7}$	0.18	2
$\;3\times 10^{3}\;V/m\;$	248.6 THz	226.1 THz	2.2 $\times 10^{7}\;$	0.19	4
$\;5\times 10^{3}\;V/m\;$	243.8 THz	258.5 THz	2.7 $\times 10^{7}\;$	0.18	6
$\;7\times 10^{3}\;V/m\;$	234.3 THz	288.7 THz	2.3 $\times 10^{7}\;$	0.18	9
1.3 $\times 10^{4}\;V/m\;$	249.8 THz	341.4 THz	3.5 $\times 10^{7}\;$	0.18	12
3.4 $\times 10^{4}\;V/m\;$	259.9 THz	331.9 THz	5.2 $\times 10^{7}\;$	0.21	15
2.2 $\times 10^{5}\;V/m$	272.8 THz	363.6 THz	4.6 $\times 10^{7}\;$	0.21	18
9.4 $\times 10^{5}\;V/m\;$	277.7 THz	411.1 THz	3.9 $\times 10^{7}\;$	0.18	21
4.3 $\times 10^{6}\;V/m\;$	270.4 THz	448.2 THz	6.5 $\times 10^{7}\;$	0.18	24
1.1 $\times 10^{7}\;V/m\;$	261.5 THz	475.3 THz	8.8 $\times 10^{7}\;$	0.20	27
2.7 $\times 10^{7}\;V/m\;$	265.3 THz	475.8 THz	7.1 $\times 10^{7}\;$	0.22	30
9.1 $\times 10^{7}\;V/m\;$	266.2 THz	476.5 THz	$\;7.7\times 10^{7}$	0.22	33

**Table 3 micromachines-11-00686-t003:** Comparison of the numerical and experimental results for different excitation amplitudes with the estimated nonlinearity coefficient of $d_{est}=1.21\times 10^{-21}$ C/V.

Excitation Wave Amplitude (V/m)	Theoretical Efficiency	Numerical Efficiency	Error Percentage
$\;1\times 10^{8}\;$	$1.93\times 10^{-5}\;$	$1.92\times 10^{-5}\;$	0.5
$\;3\times 10^{8}\;$	$1.74\times 10^{-4}\;$	$1.74\times 10^{-4}\;$	0.4
$\;5\times 10^{8}\;$	$4.82\times 10^{-4}\;$	$4.88\times 10^{-4}\;$	1.2
$\;1\times 10^{9}\;$	$1.93\times 10^{-3}\;$	$1.96\times 10^{-3}\;$	1.55
$\;1.5\times 10^{9}\;$	$4.33\times 10^{-3}\;$	$4.39\times 10^{-3}\;$	1.38
$\;2\times 10^{9}\;$	$7.68\times 10^{-3}\;$	$7.76\times 10^{-3}\;$	1.04
$\;2.5\times 10^{9}\;$	$1.20\times 10^{-2}\;$	$1.21\times 10^{-2}\;$	0.83

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
