# Peer review of "High-Fidelity Harmonic Generation in Optical Micro-Resonators Using BFGS Algorithm"

_micromachines, 2020, doi:10.3390/mi11070686_

Round 1
Reviewer 1 Report
Manuscrit "Microscale Ultra-Efficient Harmonic Generation via Non-Linear Programming of Optical Micro-Resonator Excitations" by the authors Özüm Emre Aşırım and Mustafa Kuzuoğlu.
In the proposed work authors investigate the ultra-high efficiency microscale optical resonators to be used for generating super-intense harmonics. Work appears of interests and I suggest to be accepted for publication in Micromachines after minor revisions.
- captions of figure 1,3 and 7 could be extended for a better understanding of the different figures
- In figure 2, It is unclear to me the purpose of the diagonal arrow, could you please clarify the figure.
Reviewer 2 Report
This work reports a nonlinear programming approach to achieve an ultraefficient harmonic generation at a desired optical frequency. This article proved that generating intense harmonic could be done using ultra-high efficiency microscale optical resonators for enhancing the efficiency of integrated photonic devices.
Major comments:
1: The abstract is too confusing. It took the authors four sentences to introduce the previous reason why they want to study in this article. I recommend rewriting the abstract to make it concise and on target.
2: The authors mentioned the damping ratio of each mode, but they didn't specify the value of each damping ratio. Did you assume all the damping ratio equal? It is hard to imagine the same damping ratio to all modes.
3: Figure 3: The reviewer is not sure why the authors show two half-sine waves. The figure doesn't add any information; instead makes it confusing—the same issue for figure 7.
4: Figure 5: Please use a log scale for both axes (also for any other spectrum figure). Also, please add the units for the spectral magnitude.
5: Figure 6: what are the specification of the filter used? (Q factor). Also, the damping ratio is essential to figure out how the authors got the figure.
6: Also, please use the log scale for figures 9 and 10.
Minor comments:
1: The title is too long.
2: These references are not sufficient. There are many groups dealt with issues regarding nonlinear and mixing frequencies on microresonators but not optical (Younis group and Krylov group). Still, I believe the mixing of frequencies is not a novel approach.
Here are some of them
1. Hasan, M. H., Alsaleem, F., & Ramini, A (2019). Voltage and deflection amplification via double resonance excitation in a cantilever microstructure. Sensors (Switzerland), 19(2). ISBN/ISSN #/Case #/DOI #: 14248220
2. Ramini, A., Ibrahim, A. I., & Younis, M. I. (2016). Mixed frequency excitation of an electrostatically actuated resonator. Microsystem Technologies, 22(8), 1967-1974. ISBN/ISSN #/Case #/DOI #: 09467076
3. Jaber, N., Ramini, A., & Younis, M. I. (2016). Multifrequency excitation of a clamped–clamped microbeam: analytical and experimental investigation. Microsystems and Nanoengineering, 2.
4. Shi, Yu, Wonseok Shin, and Shanhui Fan. "Multi-frequency finite-difference frequencydomain algorithm for active nanophotonic device simulations." Optica 3, no. 11 (2016): 1256-1259.
5. Ji, Hongyu, Bo Zhang, Guocui Wang, Wei Wang, and Jingling Shen. "Photo-excited multifrequency terahertz switch based on a composite metamaterial structure." Optics Communications 412 (2018): 37-40.
6. Asfaw, A. T., A. J. Sigillito, A. M. Tyryshkin, T. Schenkel, and Stephen Aplin Lyon. "Multifrequency spin manipulation using rapidly tunable superconducting coplanar waveguide microresonators." Applied Physics Letters 111, no. 3 (2017): 032601.
There are many new articles dealing with mixing and control on the micro and nanoscale.
3: "Multi-resonant" instead "multiresoant" (also adjust throughout the manuscript.
4: Line 115 "indicated" instead of "indiated".
